**Subject Area:**
genetics

cancer genetics, evolutionary biology, mathematical modelling

**Authors for correspondence:**
Xiaowei Jiang
e-mail: xiaowei.jiang@xjtlu.edu.cn
Ian P. M. Tomlinson
e-mail: ian.tomlinson@igmm.ed.ac.uk

[†]Present address: Department of Biological Sciences, Xi'an Jiaotong-Liverpool University, 111 Ren'ai Road, Suzhou 215123, People's Republic of China

# Why is cancer not more common? A changing microenvironment may help to explain why, and suggests strategies for anti-cancer therapy

Xiaowei Jiang[†] and Ian P. M. Tomlinson

Edinburgh Cancer Centre, MRC Institute of Genetics and Molecular Medicine, University of Edinburgh, Crewe Road South, Edinburgh EH4 2XU, UK

(iD) XJ, 0000-0002-6030-1249

One of the great unsolved puzzles in cancer biology is not why cancers occur, but rather explaining why so few cancers occur compared with the theoretical number that could occur, given the number of progenitor cells in the body and the normal mutation rate. We hypothesized that a contributory explanation is that the tumour microenvironment (TME) is not fixed due to factors such as immune cell infiltration, and that this could impair the ability of neoplastic cells to retain a high enough fitness to become a cancer. The TME has implicitly been assumed to be static in most cancer evolution models, and we therefore developed a mathematical model of spatial cancer evolution assuming that the TME, and thus the optimum cancer phenotype, changes over time. Based on simulations, we show how cancer cell populations adapt to diverse changing TME conditions and fitness landscapes. Compared with static TMEs, which generate neutral dynamics, changing TMEs lead to complex adaptations with characteristic spatio-temporal heterogeneity involving variable fitness effects of driver mutations, subclonal mixing, subclonal competition and phylogeny patterns. In many cases, cancer cell populations fail to grow or undergo spontaneous regression, and even extinction. Our analyses predict that cancer evolution in a changing TME is challenging, and can help to explain why cancer is neither inevitable nor as common as expected. Should cancer driver mutations with effects dependent of the TME exist, they are likely to be selected. Anti-cancer prevention and treatment strategies based on changing the TME are feasible and potentially effective.

## 1. Introduction

Although cancer is viewed as a common disease, our current knowledge of how cancers grow suggests that cancer is actually far less frequent than we expect [1]: the mutation rates and the number of normal cells suggest that many cancers could occur in each human. There are several potential reasons for the 'cancer deficiency', including anti-cancer immune responses [2] and inherent mechanisms of tissue homoeostasis (or 'buffering') [3]. We hypothesized that a further contributory factor restraining carcinogenesis is that the tumour microenvironment (TME) is not fixed, and that this could impair the ability of neoplastic cells to retain a high enough Darwinian fitness to become a cancer.

Carcinogenesis has long been viewed as an ecological and evolutionary process [4–10], epitomized by Nowell's illustrated phylogenetic model in which clonal and stepwise accumulation of advantageous mutations under selection from the TME cause a cancer to grow [5]. Although the seminal ideas proposed by Nowell and others [5,11] have guided cancer research for decades, a general theory of adaptive cancer evolution is lacking [6,8–10,12]. Some fundamental questions remain unanswered, including the role of a changing TME in determining the fitness effects of new driver mutations and the evolutionary trajectories of cancers in three-

dimensional (3D) tissue space. Does cancer evolution proceed by driver mutations with large effects, small effects or a combination of both? How and why do the fitness effects of driver mutations differ across cancer types? What spatio-temporal patterns of cancer development can be observed under different tempos and modes of adaptive evolution?

Factors intrinsic to the cancer cell, such as mutations and stable epigenetic features, and the TME should jointly determine the tumour's evolutionary trajectory [1,5,13–18]. The TME can include many non-neoplastic cells and acellular components, including stromal constituents such as fibroblasts, immune cells, the extracellular matrix (ECM) and the pre-metastatic niche. These features may, in part, be caused by the cancer cell itself. The TME is thought to have a dynamic influence on tumour progression, and metastasis [1,15,19–23], spatially varying immune TME [2] and cycling hypoxic TMEs might play important roles in driving cancer evolution [24,25]. Conceptually, cancer development can be viewed as an adaptive evolution process in a changing microenvironment, where the cancer cell survives if its phenotype is close to the optimum resulting from cell-intrinsic (cell-autonomous) factors and the TME.

In previous ecological and evolutionary modelling of cancer, different aspects of cancer development and treatment, such as tumour initiation and progression, cell–cell interactions and TME effects, have been studied using a plethora of ecological, evolutionary and mathematical approaches (e.g. [26,27]). However, due to challenges in modelling the TME explicitly, the role of a changing TME in determining the fitness effects of mutations in spatially evolving cancers has not generally been considered, particularly in 3D [28,29]. When the TME has specifically been considered as a source of selection [13,30,31], the underlying genetic basis—such as the variable fitness effects of driver mutations—has not been explicitly considered. Moreover, the important question of how cancer cells adapt in a TME with a changing phenotypic optimum, for example, due to a variable immune response [18], has not generally been considered.

To address these issues within the same evolutionary and ecological framework, with the dual aims of providing a general model of adaptive cancer evolution and investigating the effects of a changing TME on cancer growth, we set up a mathematical model of carcinogenesis as an adaptive evolutionary process. Here, we give an overview of the method. To model spatially evolving cancers using fitness landscapes, we establish a phenotypic and genetic model (see Results and Methods for details). Cancer adaptation is modelled by Fisher's geometric model with random mutation and a changing phenotypic optimum. A transformed tissue stem cell is assumed to acquire a genetic or a phenotypic change that initiates cancer growth with a certain fitness that changes in each cell of the growing cancer with mutations and a changing TME. Properties of genotypic landscapes of selected driver mutations are characterized by using the concepts of Sewall Wright's genotypic fitness landscapes. The results of the simulations help to explain why cancer growth is an unpredictable, unstable and uncertain process, with implications for anti-cancer strategies.

# 2. Results

## 2.1. Description of the model

Our model, which is based on Fisher's classical phenotypic geometric model [12,32], considers a tumour that comprises multiple neoplastic cells, each of which has a phenotype determined by random mutations and a Darwinian fitness determined by how far the phenotype is from an optimum determined in part by the TME. The adaptive process Fisher described is analogous to adjusting the knobs of a microscope (mutations) and each resulting trait contributes to fitness independently (universal pleiotropy) [33]. In Fisher's original analogy, the object on the microscope's stage is static, so the population adapts to a static phenotypic optimum in that environment. In our model, we assume that the TME has a phenotypic optimum that follows various changing patterns. In each cancer cell, the changing phenotypic optimum leads to continual selection acting on the fitness effects of mutations that influence the phenotypic traits of cancer cells.

To model the above adaptive process, our model uses fitness landscapes (electronic supplementary material, figures S1 and S2), with a single changing phenotypic optimum determined by the TME, which can move directionally or otherwise (e.g. randomly or cyclically), as the phenotypic optimum changes (see Methods for model details and electronic supplementary material, figure S1). The model can also consider cell-autonomous mutations, which we assume to act independently of the TME. In a very simple landscape, the fitness of a cancer cell with two traits can be described by a function in a 3D Cartesian coordinate system, where the height along the fitness surface corresponds to fitness and the other two coordinates correspond to phenotypic values of each trait. The initial optimal phenotypic value defining the maximum fitness is at the origin of the Cartesian coordinate system, which points to the initial 'peak' of the fitness landscape (electronic supplementary material, figure S2).

Mutation rate and the number of loci are specified before each simulation. The former could, for example, be increased owing to aberrations in key pathways maintaining DNA replication fidelity, or extrinsic mutagens (electronic supplementary material, figure S1b). If an allele becomes fixed in the population, we say this is an adaptive or positively selected driver mutation, even if subsequent phenotypic optimum changes render that driver mutation disadvantageous or neutral. This does not exclude the possibility that mutations may apparently be 'selected' due to hitchhiking, even including variants with negative fitness coupled to a mutation with a stronger, positive effect. Competition between tumour subclones (or 'clonal interference') can occur as a consequence of limits we optionally set on tumour size.

## 2.2. Cancer adaptation under diverse cancer initiation conditions and a changing phenotypic optimum

In our modelling framework, we make some general, plausible assumptions for cancer initiation and how the phenotypic optimum changes, such that the neoplastic growth can be initiated by a single stem cell with any initial fitness. In some cases, we assume that the cell population has undergone a modest initial expansion analogous to a benign cancer precursor lesion [34]. We simulate cancer evolution under various initial fitness conditions and phenotypic optima.

First, we simulate cancer evolution with a static phenotypic optimum ($v_1 = 0$; see Methods). A single cell with two traits ($n = 2$) starts asexual reproduction from the centre of a 3D tumour tissue space with a phenotype at the optimum with initial fitness $w_0 = 1$. This assumption is used to simulate a stem cell with an optimum fitness conferred by a driver

royalsocietypublishing.org/journal/rsob    Open Biol. **10**: 190297

royalsocietypublishing.org/journal/rsob    Open Biol. **10**: 190297

mutation, which is ensured to form a cancer with initial cell-autonomous phenotype and growth in its local TME. The cancer starts to grow with cell birth rate $w(\mathbf{z},t)$, and death rate $1 - w(\mathbf{z},t)$. The cancer undergoes classical constant stabilizing selection, where purifying selection removes cells with deleterious mutations. The mean population fitness remains high and does not fluctuate in time or space (electronic supplementary material, movie S1). When under constant stabilizing selection, we find that there is no fixation of any new driver mutations (electronic supplementary material, figure S3 and movie S1), consistent with the 'big bang' model of tumorigenesis in colorectal cancer [29].

Once any type of changing TME—and hence changing phenotypic optimum—is imposed, several interesting patterns of evolution emerge. The fluctuating fitness of cancer cells in time and space caused by a moving phenotypic optimum and random mutations can result in complex patterns of clonal and subclonal evolution (electronic supplementary material, movies S2–S6). Different morphologies based on varying clonal or subclonal population structures can emerge (see examples in figure 1). The speed of phenotypic optimum change is critical. If the phenotypic optimum moves too fast, and the cancer cells do not acquire enough beneficial mutations to increase the mean population fitness, the population goes extinct (electronic supplementary material, movies S2–S4). However, if the optimum moves at a moderate speed, the cancer population may acquire enough beneficial mutations to 'catch up' with the phenotypic optimum (electronic supplementary material, movies S5–S6). If the optimum moves slowly, the tumour can grow almost exponentially to the maximum space and/or population size allowed ($n = 10^7$), typically forming a ball (electronic supplementary material, movies S5–S6), and then continue to evolve with few new driver mutations.

We find two general clonal or subclonal growth patterns in 3D: (i) one or more balls of cells (figure 1a–i) or (ii) irregular morphology (figure 1j–l). Intriguingly, these patterns generally resemble those observed clinically [20,28]. As a fast-changing TME often leads to fast cell turnover, we are more likely to observe ball-like structures (figure 1a–l; electronic supplementary material, movies S3–S6). Moreover, fast-changing TMEs can lead to many smaller subclones, which turn over fast, with fluctuating numbers, spatial proximity and fitness (figure 1a–c; electronic supplementary material, movie S3). We also find that subclonal populations mix frequently with variable fitness and spatio-temporal heterogeneity during evolution (figure 1; electronic supplementary material, movies S2–S6). In a relatively static or an extremely slow-changing TME, spatio-temporal clonal turnover is slow and the cancers often show an irregular morphology (e.g. figure 1g–l; electronic supplementary material, movie S6). This is because selection is quite weak and hence adaptive evolution is limited. Moreover, to understand the fitness effects of selected driver mutations in a changing TME, we measured the mean population fitness and the mean fitness advantage of selected driver mutations, which are mostly determined by the speed of phenotypic optimum change (figure 2; electronic supplementary material, movies S2–S6). As expected, the mean population fitness decreases faster in a fast-changing TME, whereas a slow-changing/static TME leads to slower fitness decay and thus longer survival time of the tumour. Lower selection intensities (a 'flatter' fitness landscape; electronic supplementary material, figure S2) can also mitigate against a changing TME,

significantly increasing the mean fitness of the cancer cell population (electronic supplementary material, figures S4–S6).

Second, we now consider that the population starts from a phenotype away from the optimum, and thus has lower initial fitness ($w_0 < 1$). If the initial fitness is relatively low (e.g. $w_0 = 0.1$), the chance of population extinction is high (figure 3; electronic supplementary material, movies S7–S11). We conclude that in this scenario, a cancer is much more likely to grow if it is initiated by a cell-autonomous driver mutation conferring relatively high fitness that has small or zero dependency on the TME. This means that relatively low fitness cells can gain higher fitness and, with a larger 'initial' population size, have a higher chance of generating appropriate driver mutations to survive selection due to either sudden or rapid phenotypic optimum change. If the population survives the initial selection, and assuming that the phenotypic effects of subsequent mutations are TME-dependent, its evolutionary trajectory will depend on the rate of phenotypic optimum change (figure 3; electronic supplementary material, movies S8–S15). Again, our simulations show three similar patterns as above. First, a fast-changing TME leads to the selection of driver mutations with large fitness advantage and quick population extinction, which, however, can be delayed if the initial population size is large (figure 3d,h,l). Second, a moderately changing TME promotes cancer adaptation by fixing more driver mutations (figure 3b,f,j; electronic supplementary material, movies S9 and S13). Third, if the population evolves under a slowly changing TME, we recover subsequent driver mutations that have small fitness effects (figure 3a,e,i; electronic supplementary material, movies S7 and S15). These mutations could be termed 'mini-drivers', as we proposed previously [35]. These results show that cancer is unlikely to reach an optimum by using major drivers alone. Further exploration of five mechanisms potentially affecting cancer adaptation—chromosome instability, mutation rate, number of cancer traits, selection correlation and cell–TME interaction—is shown in electronic supplementary material, figures S7–S10 and notes S1–S3.

Next, to understand how other types of changing phenotypic optimum affect cancer adaptation, we analyse another three types of changing TME (randomly fluctuating, directionally changing with a randomly fluctuating component and cyclic). These results show similar patterns to the directionally changing optimum, indicating that our conclusions regarding highly uncertain evolutionary trajectories caused by changing TME are robust. There are, however, some novel patterns observed when the TME changes in a non-directional fashion, such as that a cycling phenotypic optimum is particularly capable of promoting adaptive evolution by periodically fixing more driver mutations (electronic supplementary material, figures S11–S13, movies S16 and S17, and note S4). Moreover, the variance in population fitness can become lower when the frequency of the cycling TME optimum becomes higher (electronic supplementary material, figure S14). In general, a directionally changing optimum more likely causes reduced mean population fitness and thus more killing of cancer cells. The addition of random changes in the phenotypic optimum can lead to further killing of cancer cells.

Finally, we explore how the various TME dynamics shown above affect subclonal cell–cell competition and properties of driver mutations, especially their genotypic landscapes (see Methods). We show that changing TMEs can generate genotypic landscapes of considerable ruggedness and sign epistasis, meaning that predicting cancer adaptation from the

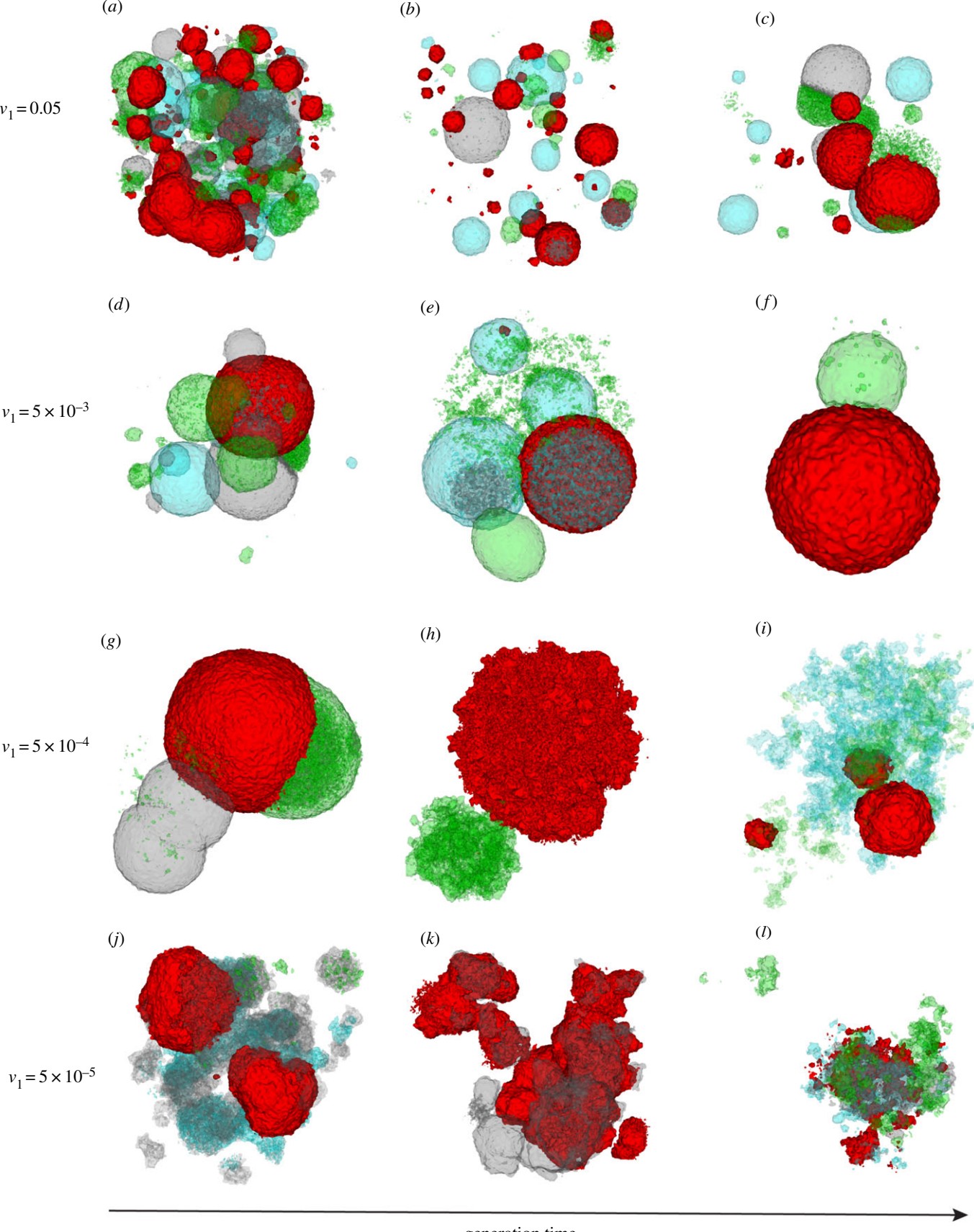

**Figure 1.** Three-dimensional simulation snapshots of cancer adaptive evolution under various directionally changing TME selective optima. Illustrative 3D snapshots of the evolving cancer cell population are taken sequentially, under different phenotypic optimum change rates, $v_1$. $(a–c)$ $v_1 = 0.05$, $(d–f)$ $v_1 = 5 \times 10^{-3}$, $(g–i)$ $v_1 = 5 \times 10^{-4}$ and $(j–l)$ $v_1 = 5 \times 10^{-5}$. To illustrate subclonal heterogeneity of fitness through time and 3D space, cancer cells are coloured according to their fitness cut-off values using quartiles, which results in four groups of cancer cells coloured with four different colours: red (the highest 25%), cyan (between the highest 25% and 50%), grey (between the lowest 50% and 25%) and green (the lowest 25%), respectively. To show 3D mixing of subclones with different fitness, we applied lower opacity to cyan, grey and green colours representing the lowest 75% cancer cell populations in fitness. Note that, because of immediate population extinction, data are not shown for simulations with phenotypic optimum change rate at $v_1 = 0.5$. The width of the fitness landscape (i.e. selection intensity) is set to $\sigma^2 = 10$ (equation (4.2); electronic supplementary material, figure S2).

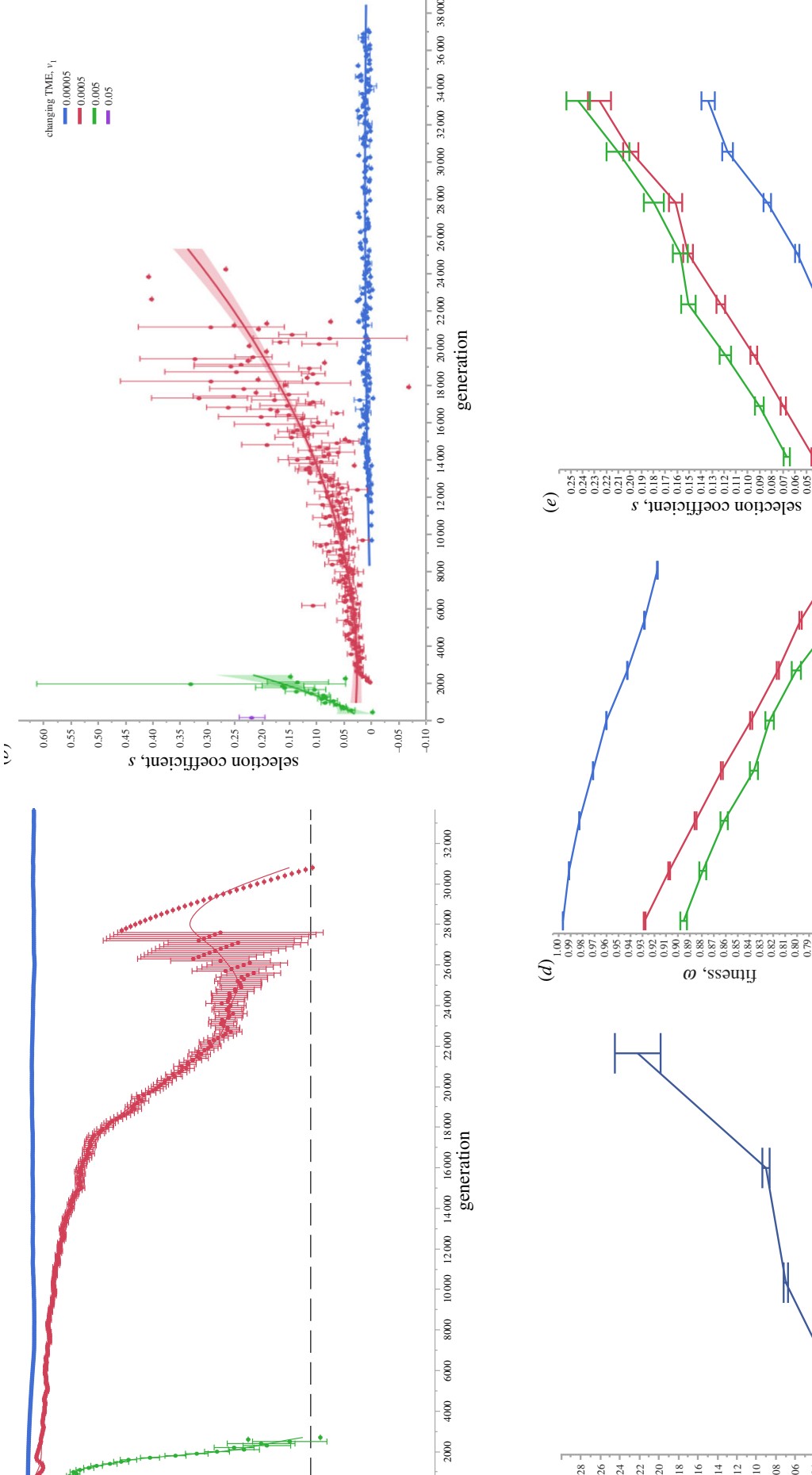

**Figure 2.** Representative cancer evolution trajectories with different rates of directionally changing TME selective optima. The dashed line represents the mean fitness 0.5. When the mean population fitness reaches this value, it is more likely to be extinct. (*a*) The mean population fitness of the evolving cancer cells is sampled at various phenotypic optimum change rates. The line represents a simple linear regression fit. (*b*) The mean selection coefficients are sampled at various phenotypic optimum change rates. The line represents a smooth curve. (*c*) The mean selection coefficients are plotted against the phenotypic optimum change rates. (*d*) The mean population fitness is plotted against the number of traits of cancer cells. (*e*) The mean selection coefficients are plotted against the number of traits of cancer cells. The TME rates used are $v_1 = 0.5$, $v_1 = 0.05$, $v_1 = 5 \times 10^{-3}$, $v_1 = 5 \times 10^{-4}$ and $v_1 = 5 \times 10^{-5}$. Note that, because of immediate population extinction, data are not shown for simulations with phenotypic optimum change rate at $v_1 = 0.5$. Error bars are the standard error of the mean (s.e.m.), and each point represents 100 independent simulations.

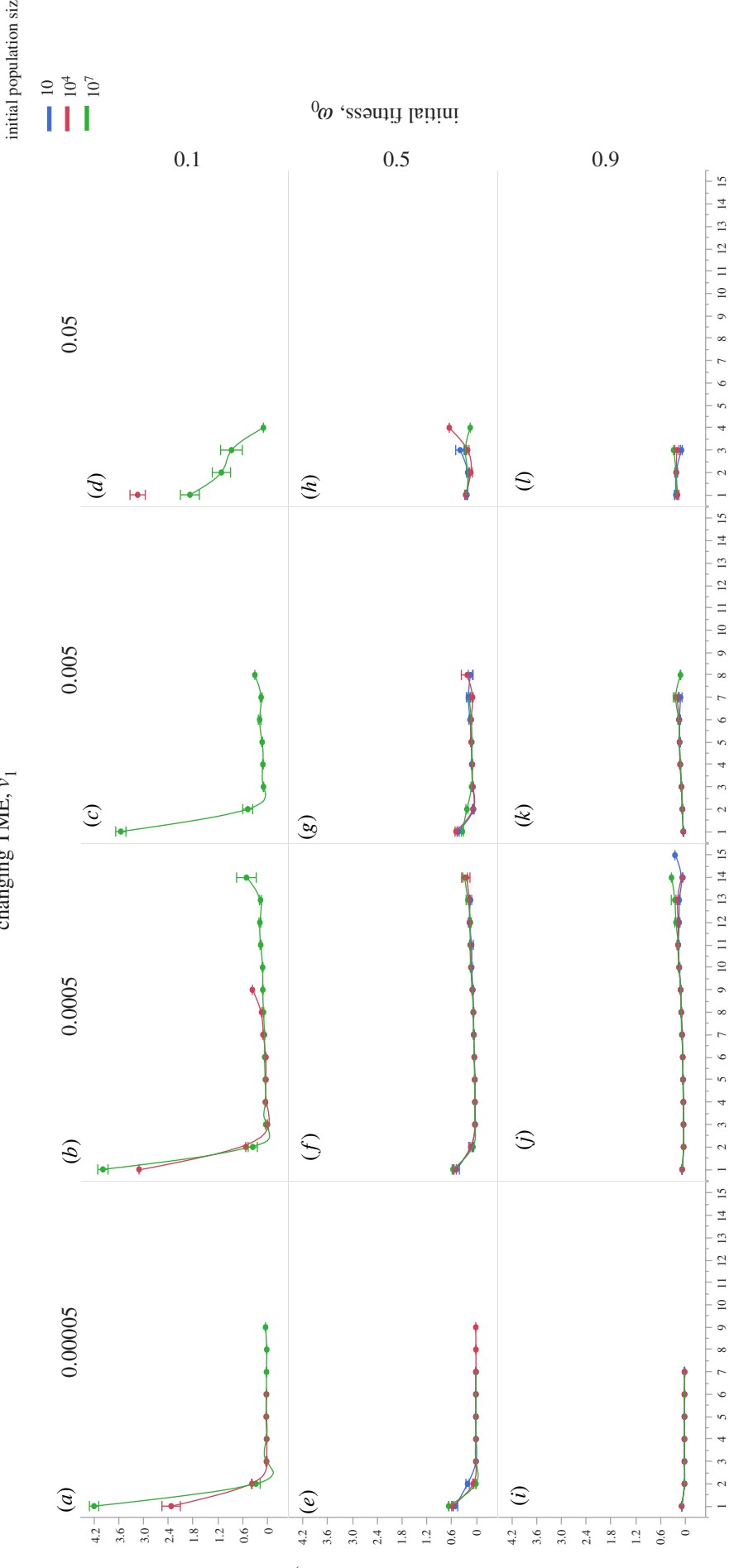

**Figure 3.** Cancer adaptive evolution under multiple different initial conditions. Simulations are performed with different initial conditions regarding population size, fitness and TME optimum change rates. Three conditions for initial population size are used, $n = 10$ (coloured blue), $n = 10^4$ (coloured red) and $n = 10^7$ (coloured green), respectively. Populations also evolve under four different TME optimum change rates: $v_1 = 5 \times 10^{-5}$ ($a,e,i$), $v_1 = 5 \times 10^{-4}$ ($b,f,j$), $v_1 = 5 \times 10^{-3}$ ($c,g,k$) and $v_1 = 0.05$ ($d,h,l$), respectively. Three conditions for initial fitness are also used, $w = 0.1$ ($a$–$d$), $w = 0.5$ ($e$–$h$) and $w = 0.9$ ($i$–$l$). Note that in ($e$–$l$), all colour lines are present, but they are too close to each other to be seen clearly. Error bars are s.e.m. and each point represents 100 independent simulations.

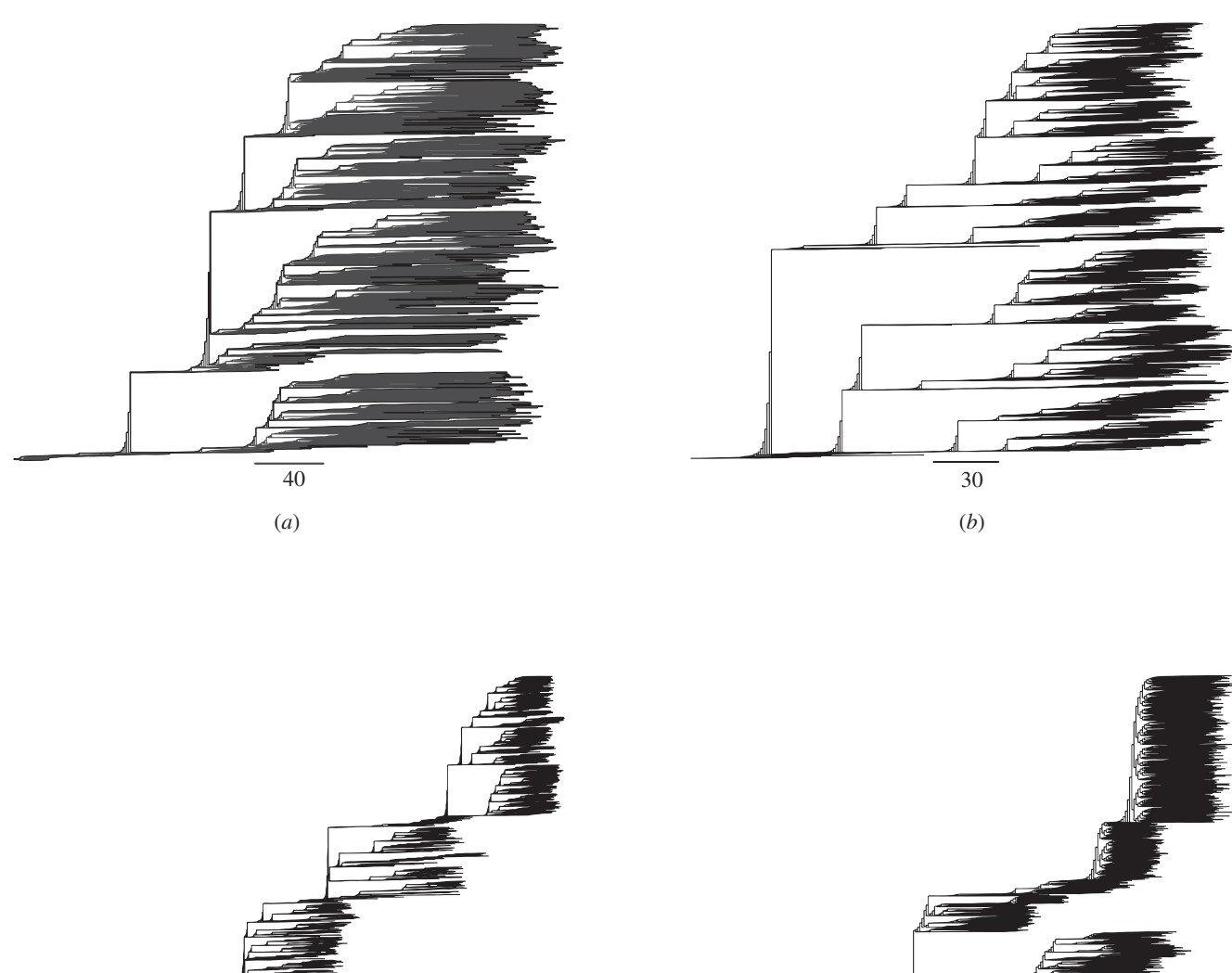

**Figure 4.** Cancer phylogenies under directionally changing TME selection dynamics. These results demonstrate that it is possible to use cancer phylogenies to understand the underlying TME selection dynamics due to either cancer treatments or somatic evolution. Example phylogenies are shown for simulated cancers under three different TME selective dynamics. (*a–d*) Four illustrative phylogenies are shown for a directionally changing TME optimum with four different speeds: $v_1 = 0$ (static TME) (*a*), $v_1 = 5 \times 10^{-5}$ (*b*), $v_1 = 5 \times 10^{-4}$ (*c*) and $v_1 = 5 \times 10^{-3}$ (*d*). All cancers were longitudinally sampled for every 100 generations for a fixed period of 10 000 generations except (*d*), where the cancer went extinct at approximately 2300 generations. The maximum population size is set at $n = 10^5$. The scale bar represents the number of cell divisions.

complement of driver mutations may be misleading in this scenario. Further explorations of subclonal cell–cell competition and evolutionary dynamics under a changing TME are shown in electronic supplementary material, figures S15–S25, movies S18–S21, and notes S5 and S6.

## 2.3. Both a changing phenotypic optimum and spatial constraints on population size affect cancer phylodynamics and adaptation

To understand the effects on cancer phylodynamics of variable selection caused by a changing TME (see Methods), we reconstruct cancer phylogenies in the simulations shown above

within a 3D TME space of the same size. Intriguingly, we find that in all cases, the shapes and the temporal signals in the phylogenies (e.g. the local branch length and overall shape) are characteristic of the optimum change dynamics, that is, TMEs with the phenotypic optimum changing at different speeds (illustrated in figure 4). In all cases, phylogenies from a slowly changing TME have long parallel branches indicating subclones may coexist for long periods of time without fixation (figure 4*a,b*), suggesting weak selection on subclones. A moderately changing optimum (figure 4*c*) typically leads to the continual selection of branches with beneficial mutations and thus favours adaptive evolution. A fast-changing phenotypic optimum causes strong selection and shorter branches indicating fast extinction of the subclones (figure 4*d*). Similar

results are obtained for other optimum change dynamics (illustrated in electronic supplementary material, figure S26).

As the available cancer niche space may also affect cancer adaptation and phylodynamics, for example, eliminating less fit cancer cells, we repeat the simulations shown above using the three (directional, cyclic and random) optimum change dynamics with different spatial constraints on cancer size and reconstructed all phylogenies during the simulations. We further use two measures for quantifying asymmetry of these phylogenies, the normalized Sackin's index and the number of cherries (see Methods), respectively. We find that both spatial constraints and a changing phenotypic optimum can affect cancer adaptation and phylodynamics to various degrees (electronic supplementary material, figures S27 and S28, and note S7). Particularly, when the changing TME causes strong selective pressure smaller 3D space hinders adaptive evolution while larger 3D space promotes it.

## 2.4. Cancer adaptation owing to anti-cancer therapies

Given our finding that a changing TME tends to promote cancer extinction or suboptimal growth, we now extend our model to simulate the effects of different anti-cancer therapeutic strategies, by assuming that these modify the phenotypic optimum and/or the shape of the fitness landscape (see equation (4.17); electronic supplementary material, figure S2). We examine the effects of treatments such as genotoxic therapies (radiotherapy or chemotherapy) [36] and immunotherapy [2] by assuming that they can cause a sudden shift in the phenotypic optimum and hence place cancer cells under strong selective pressure. For simplicity, we assume that the therapy causes the optimum to move suddenly to various degrees along the axis of the first trait while keeping the optimum constant at the rest (electronic supplementary material, figure S29), our intention being to determine qualitatively what happens with a treatment-induced phenotypic optimum shift. We confirm that although all treatments assumed to be effective are successful in reducing mean population fitness and tumour mass to the brink of extinction, only treatments that induce a large sudden change of the phenotypic optimum can lead to a cure (electronic supplementary material, figures S29 and S30 and movies S22–S25). Moreover, the complementary phylogenies show that selection from the treatment-induced optimum shifts often leads to expansion of subclones and extinction of the majority of the tumour mass (electronic supplementary material, figure S30e,j,o). Weak treatment leads to a large number of initial subclones with de novo resistance mutations, but eventually the tumour mass comes to be dominated by a single subclone (electronic supplementary material, figure S30e) carrying the mutation with the largest selective advantage. Stronger, but non-lethal, treatment leads to an early subclonal expansion carrying the mutation with a very large selective advantage ($s = 26.563$; electronic supplementary material, figure S30o).

In terms of subclonal diversity, we observe a reduction soon after treatment and then an increase, but after subclonal fitness recovers to the same or similar levels as those at pre-treatment, the diversity reduces again until the end of the treatment (data not shown). These behaviours are consistent with a mean population fitness decrease due to treatment and selection for resistant subclones (electronic supplementary material, figure S30). Interestingly, our simulations show that most resistance mutations are born early but not pre-existing. In rare cases, the few pre-existing resistance mutations are deleterious, suggesting they moved these mutant subclones' phenotypes away from the original optimum and the treatments helped those subclones to survive by moving their phenotype closer to the new treatment-induced optimum. Moreover, when the treatments lead to a sufficiently high selection intensity (a 'narrower' fitness peak; see electronic supplementary material, figure S2), all treatments lead to immediate cure (population extinction).

To understand potential therapeutic resistance, we examine three mechanisms that cancers could use to increase the mean population fitness and avoid extinction (electronic supplementary material, movies S26–S28 and figure S31). First, we assume a polygenic model of resistance, which has quantitative traits. Somatic copy number variation may increase the number of genetic loci contributing to resistance, which helps the population avoid extinction and selects for resistance driver mutations ($L = 50$; electronic supplementary material, figure S31a). Second, with an elevated mutation rate—perhaps caused by the therapy itself—mean population fitness can quickly rebound and extinction is avoided, where positively selected resistance mutations arise during the simulation (electronic supplementary material, figure S31b). Interestingly, all positively selected resistance mutations have a de novo origin after treatment initiation, because all other mutations in the subclones are removed by therapeutic selection or failed to hitchhike within the dominant subclone. Third, if selection intensity decreases—for example, the therapy intensity is reduced owing to toxicity—the cancer cell population can also avoid extinction ($\sigma^2 = 40$; electronic supplementary material, figure S31c; the fitness landscape shape change is illustrated in electronic supplementary material, figure S31). Nevertheless, due to weaker selection, no positively selected mutations are detected in these simulations. In all cases, the mean cancer cell population fitness is reduced significantly if the treatment is effective. The cancer is killed off if there are no resistant subclones born early after treatment. The complex growth patterns of the resistant subclones are striking, including mixing and fast turnover (electronic supplementary material, figure S31d and movies S22–S24).

Combining the findings from our general model of cancer evolution with the models of the evolutionary responses to anti-cancer therapy suggests ways to improve treatment strategies. Treatments, such as the sequential and/or cyclical use of different therapies, that lead to a faster-moving phenotypic optimum (in any direction) may be more effective in killing cancer cells and help to reduce toxicity through the use of lower doses, as there is no requirement for initial maximum dosing to induce a large change in the selective optimum (electronic supplementary material, figures S30 and S32). The strategy is theoretically as effective as the classical maximum dosing in reducing the mean cancer cell population fitness (similar to electronic supplementary material, movies S2–S4 and figure S32), although it takes longer to work. Dosing strategies should, in principle, be optimized individually depending on the maximum allowed dose tolerance and the type of cancer (see electronic supplementary material, figure S11–S13 and movies S2–S4). In summary, although there are challenges to overcome in design and implementation, anti-cancer strategies based on moving the selective optimum appear to be highly desirable.

royalsocietypublishing.org/journal/rsob    Open Biol. **10**: 190297

# 3. Discussion

We have used mathematical models and simulations to show that a changing TME of any type is highly likely to provide a barrier to carcinogenesis, especially in the early stages of growth. Faster-changing TMEs are more effective. Rapid sub-clonal shifts and 'spontaneous' regression or tumour extinction may occur. There are also likely to be unobservable failed evolutionary trajectories of cancer growth and progression in human patients [1]. Our model may thus help to explain why cancer is not more common. In theory, a cancer could grow using only TME-independent driver mutations, but the extent to which this is possible is unclear. There is evidence that even major and/or cancer-initiating driver mutations (e.g. *APC*, *IDH1/2*, *KRAS*, *PIK3CA*, etc.) are differentially selected depending on cancer type [37], strongly suggesting a role for the TME, broadly defined, in selecting many of the common driver mutations.

Our framework does not explicitly model the TME. Rather, the TME is assumed to have a 3D space and a phenotypic optimum for cancer cells to adapt to, which can follow different changing dynamics and 3D space sizes as we have shown in this study. TMEs are highly complex in reality and challenging to model explicitly [15,26]. Therefore, this level of abstraction in our model at this stage is necessary to make advances, as this allows us to look for general patterns of cancer adaptation. Although simplistic, these phenotypic optimum change dynamics are implicated in previous studies (e.g. a directionally moving phenotypic optimum may be due to ageing [38], a variable TME may be due to dynamic immune cell infiltration [18] and a cycling TME may be due to hypoxia [24]). Overall, however, studies to determine the interaction between genotype and TME in producing a cancer phenotype are at early stages, and while data showing that, say, early and late cancers have very different TMEs are plentiful, data showing how the TME of an individual tumour changes are currently very limited.

There are some potential limitations to our work that could be addressed in future studies. For example, we did not model spatial heterogeneity in the same changing TME, as might occur with, for example, heterogeneous immune cell infiltrations [17]. To address this, we can readily extend our model to a situation in which a cancer cell's TME is influenced by neighbouring cells with spatial heterogeneity using spatial networks, where tumour evolution could be further modelled in detail by graph theories associated with the network [39–41]. Although we showed a cycling phenotypic optimum in the TME, for example, due to hypoxia, may be particularly capable of promoting adaptive evolution by periodically fixing more driver mutations, the time-scales of the plausible TME fluctuations of the optimum and relative life cycle of cancer cells are not clear. Our assumption was that the time-scale (the period) of the cycling optimum is larger than the life cycle of cancer cells. Previous studies have long showed that the generation time of cell cycles varies considerably among mammalian cells [42]. There is also evidence the cycling hypoxia is rather complex with at least two dominant time-scales, which vary between hours and days [24]. It will also be interesting to investigate how the cycling phenotypic optimum in the TME could be linked to other cell physiologies affecting cancer evolutionary dynamics, such as the circadian clock [43,44]. It will be necessary to test the effects of changing TMEs in model systems, and we argue that carefully designed animal models of carcinogenesis are required for this, rather than *in vitro* TME manipulation.

Our model supports the use of therapeutic strategies that target the TME, causing the optimum to change and potentially driving the cancer cell population to extinction [1,16]. A similar approach, termed 'adaptive therapy' or 'metronomic chemotherapy', has been proposed previously [45,46], where treatments have been adjusted with or without evolutionary modelling and there is clear evidence of improved patient outcome by cyclic dosing [47]. It is challenging to identify a phenotypic optimum that the cancer cells can actually respond to, but there have been clinical trials based on this philosophy [48,49]. In the future, our fitness landscape-based spatial modelling could be extended to measure the variable fitness effects of new driver mutations in cancer cells' adaptation to changing TMEs and treatments. This could be achieved by using longitudinally sampled whole-genome sequencing data with TME information. There has been some success in measuring patient-specific subclonal selection coefficients by assuming a static TME [50].

In conclusion, our model using phenotypic and genotypic fitness landscapes with a changing phenotypic optimum provides a single unified evolutionary and ecological framework to understand adaptive cancer evolution and treatment. Our approach is heuristic and we show that the extension of Fisher's geometric model is sufficient to produce many of the observed complex cancer evolutionary and pathology patterns. Although the phenotypic fitness landscapes are smooth, the underlying rugged genotypic landscapes of selected driver mutations due to a changing TME suggest that it is challenging to predict cancer evolution using genotypic data alone. Our focus in this study on the effect of a changing TME shows that cancers may not grow because they cannot adapt quickly enough to moving fitness optima, even if the TME changes are cyclical rather than unidirectional. In the future, our modelling framework could be used to measure heterogeneity in evolving cancers within and between patients. Although most anti-cancer therapies are based on direct killing of cancer cells, the use of therapy designed to cause a changing TME could lead to death through natural selection.

# 4. Methods

## 4.1. Fitness landscape model

In molecular evolution, organisms can evolve by positive Darwinian selection or genetic drift, or a combination of both, depending on how natural selection acts on the phenotypic traits and their plasticity [51]. The fitness advantage to organisms conferred by these phenotypic traits due to mutation and environmental selection can lead to adaptation [32,52,53]. In this study, we will use the metaphor of fitness landscape to understanding cancer adaptation, first introduced by Wright [52], which is defined by a set of genotypes, their mutational distance and fitness. As our focus is to build a phenotypic and genetic model of cancer adaptation, we first extend Fisher's geometric model of phenotypic fitness landscape to address the question of how cancer cell populations adapt to a changing TME [12,33,54–56] in 3D tissue space, which we assume to have a phenotypic optimum the cancer cells can adapt to (note cancer cells and cancer stem cells are used interchangeably). Particularly, we extend the original Fisher's

royalsocietypublishing.org/journal/rsob    Open Biol. **10**: 190297

royalsocietypublishing.org/journal/rsob    Open Biol. **10**: 190297

phenotypic landscape model into a general form by incorporating a moving phenotypic optimum with various changing dynamics (e.g. directionally, randomly or cyclically) [56–59]. Our rationale is that normal tissue homeostasis is essential for the functioning and survival of multicellular organisms, which has evolved robust buffering mechanisms to ensure that microenvironmental changes can be accommodated. Cancers may retain a few of these features, but it is implausible that they are as robust as normal tissues, given the cell types they contain and the lack of normal structures (e.g. the intestinal crypt is not recapitulated in a cancer, even if some cancers have structures reminiscent of crypts). There are thus excellent reasons to expect that a changing environment renders cancer cells more vulnerable than normal tissue. It is also important to note that our model assumes that the normal cells are buffered against (or simply not exposed to) the changing TME. There is no notion that normal cells will overgrow to out-complete the cancer—that has almost no biological basis for solid tumours. Finally, it was suggested that the ruggedness of landscapes can determine the repeatability and predictability of adaptation [60]. We will generate and characterize the underlying Sewall Wright's genotypic landscapes of selected driver mutations from our modelling framework to explore whether such genotypic landscapes are smooth or rugged.

In Fisher's geometric framework, a cancer stem cell adapting in a TME can be viewed as a point in an $n$-dimensional Euclidean phenotype space with $n$ quantitative phenotypic traits defined by a column vector $\mathbf{z} = (z_1, \ldots, z_n)^{\mathrm{T}}$. The traits involved can be any, but could for convenience be those highlighted by Hanahan & Weinberg [19]: sustaining proliferative signalling, evading growth suppressors, resisting cell death, enabling replicative immortality, angiogenesis, activating invasion and metastasis, reprogramming of energy metabolism and evading immune destruction [19]. As we assume that the TME is the primary source of selection [1,6,14,58,61–63], there is a corresponding optimum phenotype $\mathbf{z}^{\mathrm{opt}}$, which is defined by a column vector of $n$ values $\mathbf{z}^{\mathrm{opt}} = (z_1^{\mathrm{opt}}, \ldots, z_n^{\mathrm{opt}})^{\mathrm{T}}$. The Euclidean distance, $d$, is defined as $d = \|\mathbf{z} - \mathbf{z}^{\mathrm{opt}}\| = \sqrt{\sum_{k=1}^{n} (z_k - z_k^{\mathrm{opt}})^2}$. The phenotypic fitness function for a cancer stem cell in a microenvironment is defined as $w(d) = \exp(-ad^2) = \exp(-a\|\mathbf{z} - \mathbf{z}^{\mathrm{opt}}\|^2)$ [56]. Here, $a$ is the selection intensity for all traits ($a > 0$) [56]. Therefore, the fitness of an individual cancer stem cell depends on its phenotype's Euclidean distance to the optimum. The change of phenotypic traits of a cancer stem cell due to random mutations can be defined as an $n$-dimensional random number $\mathbf{r} = (r_1, \ldots, r_n)^{\mathrm{T}}$. The size of a mutation (the effect of a mutation on phenotypic traits) is therefore defined as $\|\mathbf{r}\| = \sqrt{r_1^2 + r_2^2 + \cdots + r_n^2}$. So, the combined phenotypic trait of a cancer stem cell $\mathbf{z}'$ with mutation $\mathbf{r}$, relative to its wild-type $\mathbf{z}$ is defined as $\mathbf{z}' = \mathbf{z} + \mathbf{r}$. We assume the phenotypic effect of a mutation with size $\mathbf{r}$ follows a multivariate normal distribution with mean $\mathbf{0}$ and $n \times n$ variance–covariance matrix $\mathbf{M}$ [55,64]. The selection coefficient of a mutation that changes the fitness of the cancer stem cell is therefore defined as

$$s \equiv \frac{w(\mathbf{z}')}{w(\mathbf{z})} - 1 = \frac{w(\mathbf{z} + \mathbf{r})}{w(\mathbf{z})} - 1; \tag{4.1}$$

when $s > 0$, the mutation is beneficial, as it moves the cancer stem cell closer to the optimum. When the cancer stem cell is

at optimum, we say it has its maximum fitness $w(\mathbf{z}) = 1$, whereas when the cancer stem cell moves away from the optimum, its fitness decreases ($0 \leq w(\mathbf{z}) < 1$). Such mutations are deleterious and lead to negative selection coefficients with $s < 0$. When mutations do not change fitness, they are defined as neutral mutations with $s = 0$.

We can now define a general form of the fitness function as shown before [33,54,55,65];

$$w(\mathbf{z}, t) = \exp\left[-(\mathbf{z} - \mathbf{z}^{\mathrm{opt}}(t))^{\mathrm{T}} \mathbf{S}^{-1} (\mathbf{z} - \mathbf{z}^{\mathrm{opt}}(t))\right], \tag{4.2}$$

where $\mathbf{S}$ is a real $n \times n$ positive definite and symmetrical matrix, and T denotes transposition. The matrix $\mathbf{S}$ describes the shape of the fitness landscape, namely, the selection intensity. Matrix $\mathbf{S}^{-1}$ is the inverse of $\mathbf{S}$. As introduced above, the overlapping of pathways responsible for cancer stem cell traits indicates pleiotropic effect of mutations contributing to cancer adaptive evolution [19]. If the selection intensity is the same along all $n$ traits then we have an isotropic fitness landscape (universal pleiotropy). We set $\mathbf{S} = \sigma^2 \mathbf{I}$ ($\sigma^2 > 0$, $\mathbf{I}$ is an identity matrix). Selection may also vary along different traits (selection is correlated), which means mutational effects contribute to fitness differently for different traits. If we set selection intensity to vary along $n$ traits, then $\mathbf{S}$ has non-zero off-diagonal entries. We illustrate different shapes of the fitness landscape (independent and correlated selection) in electronic supplementary material, figure S2. We can use $\bar{\sigma}^2 = \sqrt[n]{\det(\mathbf{S})}$ to measure the average width of the fitness landscape, where $\bar{\sigma}^2$ is defined as the geometric mean of the eigenvalues of $\mathbf{S}$. The selection coefficient of a mutation in equation (4.1) can now be defined as

$$s(\mathbf{z}, \mathbf{z}', t) \equiv \frac{w(\mathbf{z}', t)}{w(\mathbf{z}, t)} - 1. \tag{4.3}$$

We now describe how the simulation works to produce the 3D tumour cell structures using the fitness function $w(\mathbf{z}, t)$ (equation (4.2); electronic supplementary material, figures S1 and S2). The simulation is fully individual-based, following simple growth mechanics described previously [28,66]. Briefly, the offspring of newly divided cells stochastically look for available positions in a 3D lattice space. The population evolves in a discrete and non-overlapping manner. The empty 3D tumour space represents the TME that may change, following the dynamics described below, and the position in the 3D space does not affect the TME or selection (except in the model extension of cell–TME interaction; see electronic supplementary material, note S3). A computer simulation of cancer evolution proceeds in the following fashion. At each generation, the initial fitness ($w_0 = w(\mathbf{z}_0)$) of each cancer cell is conferred by a starting phenotype, $\mathbf{z}_0$, which is unchanged from the previous generation unless a new mutation (advantageous or deleterious) occurs by chance, or phenotypic plasticity arises, for example, from disturbances such as inflammation [15]. The phenotypic effect (size) of a mutation is sampled from a multivariate normal distribution. A mutation with large fitness effect can move the cancer cell a long phenotypic distance relative to its phenotypic optimum in the fitness landscape. The changing phenotypic optimum of the TME can also change each cancer cell's fitness. When the cell phenotype and/or phenotypic optimum changes, the Euclidean distance, $d$, of the phenotype from the optimum will change to $d'$, with an associated fitness value $w(d')$. All these changes in fitness

naturally lead to different levels of selection and adaptation—a cancer cell acquires higher fitness when mutations move its phenotype closer to the optimum. Depending on the fitness effects of mutations in that cell and position of the phenotypic optimum at generation, $t$, we assume that every cancer cell has to survive viability selection according to its birth rate (the probability to give daughter cells during viability selection—equivalent to its fitness at generation $t$), $w(\mathbf{z},t)$, and death rate (the probability of cell death during viability selection), $1 - w(\mathbf{z},t)$. The surviving cancer cells then reproduce asexually. New daughter cells randomly occupy available tissue space in 3D according to pre-specified limits on tumour size that may vary with time. The tumour may go extinct, persist with varying size, or grow continuously until it reaches its maximum allowed space or size (nominally representing clinical presentation), after which it continues to be under viability selection without expanding.

## 4.2. Cancer adaptation in a tumour microenvironment with a changing phenotypic optimum

In all, we consider six scenarios for how the TME's phenotypic optimum might change. Different initial population or individual cell properties (e.g. population size and fitness) during computer simulations are specified in the Results section. We assume that the TME optimum follows the multivariate-optimum model and only the first trait of $n$ traits changes [55,57,67]. It is important to note that the assumption that the fitness landscape has a single, well-defined optimum is a reasonable simplification and assumed qualitatively as suggested before (e.g. due to ageing, random tumour-infiltrating immune cells or hypoxia), which could be more complex in reality.

### 4.2.1. Directionally changing phenotypic optimum

We assume a directionally moving optimum with $\mathbf{v}t$,

$$\mathbf{z}^{\mathrm{opt}}(t) = \mathbf{v}t \tag{4.4}$$

and

$$w(\mathbf{z},\mathbf{v},t) = \exp\left[-(\mathbf{z} - \mathbf{v}t)^{\mathrm{T}}\mathbf{S}^{-1}(\mathbf{z} - \mathbf{v}t)\right], \tag{4.5}$$

where the vector $\mathbf{v} = (v_1,\ldots,v_n)'$ is the phenotypic optimum change speed. The optimum $\mathbf{z}^{\mathrm{opt}}(t)$ is time (generation) dependent and only the first trait optimum changes.

### 4.2.2. Randomly changing phenotypic optimum

Here, we assume the TME optimum of the first trait, $z_1^{\mathrm{opt}}$, changes randomly at each generation, $t$, following a normal distribution with mean 0 and standard deviation, $\delta$

$$\mathbf{z}^{\mathrm{opt}}(t) = (z_1^{\mathrm{opt}},\ldots z_n^{\mathrm{opt}}), \tag{4.6}$$

$$z_1^{\mathrm{opt}} \sim N(0,\delta^2) \tag{4.7}$$

$$\text{and} \quad w(\mathbf{z},t) = \exp\left[-(\mathbf{z} - \mathbf{z}^{\mathrm{opt}}(t))^{\mathrm{T}}\mathbf{S}^{-1}(\mathbf{z} - \mathbf{z}^{\mathrm{opt}}(t))\right]. \tag{4.8}$$

### 4.2.3. Directionally changing phenotypic optimum, with a random component

We assume a directionally moving optimum with $\mathbf{v}t$ (see equation (4.4)) and a random component $\mathbf{e}$,

$$\mathbf{z}^{\mathrm{opt}}(t) = \mathbf{v}t + \mathbf{e}, \tag{4.9}$$

$$\varepsilon_1 \sim N(0,\delta^2) \tag{4.10}$$

$$\text{and} \quad w(\mathbf{z},\mathbf{v},\mathbf{e},t) = \exp\left[-(\mathbf{z} - \mathbf{z}^{\mathrm{opt}}(t))^{\mathrm{T}}\mathbf{S}^{-1}(\mathbf{z} - \mathbf{z}^{\mathrm{opt}}(t))\right], \tag{4.11}$$

where $\mathbf{v} = (v_1,\ldots,v_n)'$ is the vector of the phenotypic optimum change and it is time dependent. The random vector $\mathbf{e} = (\varepsilon_1,\ldots,\varepsilon_n)'$ has $\varepsilon_1$ following a random normal distribution with mean 0 and standard deviation $\delta$.

### 4.2.4. Cyclically changing phenotypic optimum

In this scenario, we assume the TME optimum changes cyclically or periodically at each generation, $t$, following a general periodic function [57,64,68]

$$\mathbf{z}^{\mathrm{opt}}(t) = (z_1^{\mathrm{opt}}(t),\ldots z_n^{\mathrm{opt}}(t)), \tag{4.12}$$

$$z_1^{\mathrm{opt}}(t) = \frac{A}{2}\left[1 + \sin\left(\frac{2\pi t}{P} - \frac{\pi}{2}\right)\right] \tag{4.13}$$

$$\text{and} \quad w(\mathbf{z},t) = \exp\left[-(\mathbf{z} - \mathbf{z}^{\mathrm{opt}}(t))^{\mathrm{T}}\mathbf{S}^{-1}(\mathbf{z} - \mathbf{z}^{\mathrm{opt}}(t))\right], \tag{4.14}$$

where $A$ is the amplitude of TME optimum oscillation and $P$ is the period (the number of generations) of the TME cycle. Equation (4.13) allows the TME optimum to change periodically from 0 to $A$ and then to 0 in $P$ generations. In all simulations of this scenario, $P$ is fixed at 360 generations.

### 4.2.5. Stable (i.e. constant) phenotypic optimum

We assume the TME optimum is constant with $\mathbf{z}_0^{\mathrm{opt}} = (z_1^0,\ldots,z_n^0)'$

$$\mathbf{z}^{\mathrm{opt}}(t) = \mathbf{z}_0^{\mathrm{opt}} \tag{4.15}$$

and

$$w(\mathbf{z}) = \exp\left[-(\mathbf{z} - \mathbf{z}_0^{\mathrm{opt}})^{\mathrm{T}}\mathbf{S}^{-1}(\mathbf{z} - \mathbf{z}_0^{\mathrm{opt}})\right]. \tag{4.16}$$

### 4.2.6. Sudden change in the phenotypic optimum

We assume the TME selective optimum suddenly changes from $\mathbf{z}_0^{\mathrm{opt}}$ to $\mathbf{z}_1^{\mathrm{opt}}$

$$w(\mathbf{z},t) = \begin{cases} \exp\left[-(\mathbf{z} - \mathbf{z}_0^{\mathrm{opt}})^{\mathrm{T}}\mathbf{S}^{-1}(\mathbf{z} - \mathbf{z}_0^{\mathrm{opt}})\right], & 0 \leq t < \tau \\ \exp\left[-(\mathbf{z} - \mathbf{z}_1^{\mathrm{opt}})^{\mathrm{T}}\mathbf{S}^{-1}(\mathbf{z} - \mathbf{z}_1^{\mathrm{opt}})\right], & t \geq \tau \end{cases}. \tag{4.17}$$

In this scenario, the cancer cells evolve under constant stabilizing selection of the TME with optimum vector for $\tau$ generations. After that, the TME optimum changes suddenly with a new optimum vector $\mathbf{z}_1^{\mathrm{opt}} = (z_1^1,\ldots,z_n^1)'$ and then the cancer cells evolve under another constant stabilizing selection. For simplicity only, the optimum of the first trait changes from $z_1^0 = z_0^{\mathrm{opt}}$ to $z_1^1 = z_1^{\mathrm{opt}}$ and the rest trait values are kept 0. We set our initial TME optimum at the origin, so we have vectors $\mathbf{z}_0^{\mathrm{opt}} = (z_0^{\mathrm{opt}},\ldots,0)'$ and $\mathbf{z}_1^{\mathrm{opt}} = (z_1^{\mathrm{opt}},\ldots,0)'$ with $z_0^{\mathrm{opt}} = 0$.

royalsocietypublishing.org/journal/rsob    Open Biol. **10**: 190297

## 4.3. Simulation of cancer evolution in the three-dimensional space

We simulate the cancer cell population in 3D space in one of the six environments described above (electronic supplementary material, figure S1a). Our intention is to identify key cancer evolutionary patterns and investigate how these are influenced by model parameters, such as the number of phenotypic traits $n$ (the complexity of cancer cells), changing microenvironment (properties of the TME), variable selection among traits, population size, mutation rate, initial phenotype of cancer cells (distance to optimum), initial population size and chromosome instability affect cancer adaptation (tumorigenesis) in an ecological and evolutionary framework.

Simulations are performed under particular phenotypic optimum change patterns due to either natural cause or anti-cancer therapies as described in equations (4.4)–(4.17). Here, we assume cancer cells require at least one adaptive trait for survival (e.g. anti-apoptotic or metabolism-related traits) as also inspired by microbial adaptive evolution [69] and mammals with very low cancer incidences [70,71]. For simplicity, we set the number of traits, $n = 2$ (the dimensionality of the fitness landscape), in each simulation unless otherwise stated. The initial population has $K$ neoplastic cells, and grows from the centre of the specified 3D coordinates in the 3D square lattice (it has 26-cell cubic neighbours, also called a 3D Moore neighbourhood) until it reaches the defined maximum population size or 3D tumour space, which is set at $1 \times 10^7$ and $300^3$ due to computational resource limitations, respectively. The initial $K$ cells have an adjustable phenotype $\mathbf{z}_0$. This determines the initial fitness in the fitness landscape, which has the optimum at the origin and an adjustable shape defined by $\mathbf{S}$ in equation (4.2) (illustrated in electronic supplementary material, figure S2). The initial phenotype can also be calculated by equation (4.2) when the predefined fitness value is known. For generality, we consider that the adjustable phenotype of initial $K$ cells is determined by either classical driver mutations or phenotypic plasticity.

The cancer cells have $c$ sets of chromosomes (ploidy) and reproduce asexually with chromosome instability (CIN). To model CIN, for simplicity, we define a rate, $r_c$, following a Bernoulli process to vary the copy of paternal or maternal chromosome passed down to daughter cells (electronic supplementary material, figure S1b). Each individual cancer cell is represented by $L$ diploid loci (for computational efficiency without loss of generality $c = 2$, $L = 5$, unless stated otherwise) with additive effect on the $n$-dimensional phenotype $\mathbf{z}$.

A multifurcating tree tracks the genealogies of the alleles generated at these loci. When only one branch of an allele tree survives, an adaptive step is recorded as per standard terminology [55]. The per locus mutation rate, $u = 4 \times 10^{-5}$, is used for per genome replication [72]. Due to the high mutation rate and large population size, multiple mutations may be generated in each generation. So, there is clonal interference in our simulations. For this reason, we only use individual-based simulations in our study. Analytical results concerning equations (4.4)–(4.17) have previously been addressed [55,56,63,73,74]. In each simulation, the population genetics parameters are recorded in computer files. Some simple statistics are displayed in real time on the computer screen including time in years with 24 h doubling time (can be adjusted if required), adaptive steps (driver mutations), generation time and population size. The reason to set the doubling time as 24 h is to allow simulation of more generations to gain more insights into the underlying evolutionary dynamics (e.g. 1 day doubling time allows roughly 37 000 generations for 100 years; however, 5 days doubling time only allows about 7400 generations). The summary statistics for population fitness and adaptive steps are sampled every 100 generations unless otherwise stated. The real-time spatial evolutionary process of cancer development is visualized in 3D. Three-dimensional snapshots and 2D movies including fitness values of all cancer cells are made directly from simulations.

## 4.4. Phylogeny reconstruction and shape measure

In our simulations, the cancer grows from single cells and each survived cell divides into two daughter cells following the branching process, so we only consider binary trees. The phylogeny or the genealogy of the evolving cancer in each simulation can be reconstructed in real time by simply removing dead cells at each sampling point. Here, we use a term called 'phylodynamics' previously coined to study the shape of viral phylogenies due to the effect of evolutionary and/or ecological processes [75], which can arise due to immune selection and weak selective or non-selective forces such as space and population dynamics. Similarly, the shape of the resulting cancer phylogenies in our simulations contains useful information of the underlying evolutionary processes. In the context of our study, we specifically use this term, cancer phylodynamics, to describe a similar phylodynamic framework where cancer phylogenies could be affected by both TME selection dynamics, space and cancer cell population dynamics. We use two commonly used measures of phylogeny asymmetry [76], namely the normalized Sackin's index and the number of cherries, to quantify to what extent each TME selection dynamic and 3D space (spatial constraint) affect phylogeny shapes. These measures can be very useful to infer the underlying cancer dynamics in clinical settings. Specifically, the normalized Sackin's index measures the total topological distance between the tips and the root of the tree and it is then averaged by the total number of tips, which can increase as the tree asymmetry increases. A cherry is formed between two tips sharing a direct common ancestral node on the tree. An asymmetric tree may have more cherries than a balanced tree in our case as changing TMEs may lead to more cherries at each sampling point. The two measures are complementary in measuring tree asymmetry as the former measures the overall asymmetry, while the latter reveals recent asymmetry at each sampling point [77]. We used R packages apTreeshape (https://cran.r-project.org/web/packages/apTreeshape/index.html) and treeImbalance (https://github.com/bdearlove/treeImbalance) to perform these measures.

## 4.5. Calculate epistasis coefficient among selected driver mutations

To further characterize the selected driver mutations and their genotypic fitness landscapes, we derive the epistasis equation between pairs of driver mutations with a moving optimum. In Fisher's fitness landscape model, based on our version of

fitness function (equation (4.2)), the epistasis coefficient of a pair of mutations, $i$ and $j$, can be defined as [78]

$$e_{ij} = \log\left(\frac{w(\mathbf{z}_{ij}, t_{ij})w(\mathbf{z}_0, 0)}{w(\mathbf{z}_i, t_i)w(\mathbf{z}_j, t_j)}\right), \tag{4.18}$$

where $w(\mathbf{z}_{ij}, t)$ is the fitness of the double-mutant cell, while $w(\mathbf{z}_i, t)$ and $w(\mathbf{z}_j, t)$ are the fitness of the individual mutant cell. The initial ancestral cell has fitness $w_0 = w(\mathbf{z}_0, 0)$ with phenotype $\mathbf{z}_0$ and the optimum is at the origin. As shown above, if we define the phenotypic effect of the two mutations as $\mathbf{r}_i$ and $\mathbf{r}_j$, respectively. Then, we have the phenotype, $\mathbf{z}_i = \mathbf{z}_0 + \mathbf{r}_i$, for the mutant cell carrying the mutation $i$ and phenotype, $\mathbf{z}_j = \mathbf{z}_0 + \mathbf{r}_j$, for the mutant cell carrying the mutation $j$. The phenotype of the double mutant is $\mathbf{z}_{ij} = \mathbf{z}_0 + \mathbf{r}_i + \mathbf{r}_j$. Similarly, the TME optima are $\mathbf{z}_i^{\text{opt}}(t)$ and $\mathbf{z}_j^{\text{opt}}(t)$ for mutant cells carrying mutations $i$ and $j$, respectively. The optimum for the double mutant can be defined as $\mathbf{z}_{ij}^{\text{opt}}(t_{ij}) = \mathbf{z}_i^{\text{opt}}(t_i) + \mathbf{z}_j^{\text{opt}}(t_j)$. The above epistasis coefficient in a changing TME can be further given as

$$e_{ij} = -2(\mathbf{r}_i - \mathbf{z}_i^{\text{opt}}(t_i))^{\text{T}}\mathbf{S}^{-1}(\mathbf{r}_j - \mathbf{z}_j^{\text{opt}}(t_j)). \tag{4.19}$$

Note that, here the epistasis is also affected by the changing TME optimum and the epistasis coefficients between pairs of mutations can be directly calculated from the simulations.

## 4.6. Properties of a genotypic landscape of selected driver mutations

The ruggedness of cancer genotypic landscapes may determine the predictability and repeatability of cancer adaptation and therefore of paramount importance for managing cancer [60]. When describing a genotypic landscape, four to five genotypes are typically used. This leads to $2^4$ or $2^5$ all possible combinations of these genotypes to form a genotypic landscape. The epistasis coefficient between two mutations can be calculated when we know the fitness effects of the mutants as shown above. For simplicity, we use four selected driver mutations to summarize the genotypic landscapes generated by the extended Fisher's fitness landscape model with a changing phenotypic optimum. We further summarize these genotypic landscapes using two commonly used summary statistics [60], namely, the fraction of sign epistasis and roughness to slope ratio [79,80]. The former defines a situation where the fitness effects of driver mutations depend on their background driver mutations. The later quantifies how well a landscape can be described by a linear model where driver mutations can additively determine their fitness. There are two types of sign epistasis: simple and reciprocal (complex) sign epistasis. In simple sign epistasis, only one of the driver mutations is deleterious with the other driver mutation, while in complex sign epistasis, both driver mutations are deleterious with the other driver mutation as its background driver mutation, which could only happen in a cycling TME in our study.

Data accessibility. The computer code and related simulation data used in this study are available upon reasonable request from the corresponding author.

Authors' contributions. X.J. and I.P.M.T. conceived the study, analysed the simulation results and wrote the manuscript. X.J. developed the model and implemented the computational framework and performed the simulations.

Competing interests. We declare we have no competing interests.

Funding. X.J. is supported by an ERC advanced grant no. (EVOCAN-340560) awarded to I.P.M.T.

Acknowledgements. We thank John Welch and Sebastian Matuszewski for helpful discussion of Fisher's geometric model, and Sebastian Matuszewski and Andrea Sottoriva for providing their publication materials. We are grateful to Shazia Irshad and members of the Tomlinson laboratory for cancer biology discussions. We also thank Martijn Koppens, Roland Arnold and Yun Feng for helpful comments of the manuscript. Computation used the Oxford Biomedical Research Computing (BMRC) facility, a joint development between the Wellcome Centre for Human Genetics and the Big Data Institute supported by Health Data Research UK and the NIHR Oxford Biomedical Research Centre. We thank Robert Esnouf and Jonathan Diprose for computing assistance.

Disclaimer. The views expressed are those of the author(s) and not necessarily those of the NHS, the NIHR or the Department of Health.

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
