## [Reviewer comments · Open Biology]

Review History

RSOB-19-0234.R0 (Original submission)

Review form: Reviewer 1

Recommendation

Accept with minor revision (please list in comments)

Do you have any ethical concerns with this paper?

No

Comments to the Author

In general I would like to congratulate you on a clear and well written paper. I do have some minor suggestions which I would like to see you address prior to publication:

1. I found it a long read to get to the actual methods of the simulation and I would have preferred to have seen that after a brief introduction and survey. Forward referencing in the first sections to results obtained from the simulations only became clear after the second read. You should consider an overview of the method in the introduction.
2. In particular, and understandably, you focus on the fitness landscape mechanism to simulate

perturbations in the TME. It would be worthwhile to clarify at the start of the methods section the steps involved in the simulation growth so that it is clear how the genetic algorithm works to produce the 3D tumor cell structures. I felt that it would be instructive to lay out the steps of how the simulation uses the fitness function $w(z)$ to produce the cell populations with a worked example.

3. The model is based on the somatic mutation model of cancer, and some recent work (Reeves, M. Q., Kandyba, E., Harris, S., Del Rosario, R., & Balmain, A. (2018). Multicolour lineage tracing reveals clonal dynamics of squamous carcinoma evolution from initiation to metastasis. *Nature Cell Biology*, 20(6), 699–709. <https://doi.org/10.1038/s41556-018-0109-0>) has highlighted the role of higher order structures, specifically cell to cell interactions. It would be interesting in the discussion section to get your views on how your model could be extended to simulate such models of cancer growth.

4. The proposition regarding the dependence of the kinetics of the TME and outcomes is fascinating. I would draw your attention to the work that has been done on network evolution (which may not at first sight seem relevant) which indicates that in a dynamic environment sudden changes are undergone by networks in their fundamental morphology (the classic reference is Bianconi, G., & Barabási, A.-L. (2001). Bose-Einstein Condensation in Complex Networks. *Physical Review Letters*, 86(24), 5632–5635.

<https://doi.org/10.1103/PhysRevLett.86.5632>). This approach has been used to model cancer evolution (see Tee, P., & Balmain, A. (2019). Critical behavior of spatial networks as a model of paracrine signaling in tumorigenesis. *Applied Network Science*, 4(1).

<https://doi.org/10.1007/s41109-019-0167-7>) It would be fascinating to investigate whether the physics of critical phenomena could throw further light on the precise conditions upon which the TME cause runaway growth.

I hope you find these suggestions helpful.

Review form: Reviewer 2

Recommendation

Reject – article is scientifically unsound

Do you have any ethical concerns with this paper?

No

Comments to the Author

Comments uploaded as part of ESM

Decision letter (RSOB-19-0234.R0)

12-Nov-2019

Dear Dr Jiang,

We are writing to inform you that your manuscript RSOB-19-0234 entitled "Why is cancer not more common? A changing microenvironment may help to explain why, and suggests strategies for anti-cancer therapy" has, in its current form, been rejected for publication in *Open Biology*.

The referees have recommended that major revisions are necessary but that the manuscript has

potential; hence, we would like to actively encourage you to revise the manuscript accordingly, and resubmit. Nevertheless, please note that this is not a provisional acceptance.

The resubmission will be treated as a new manuscript and will re-enter the review process. Every attempt will be made to use the original referees, but this cannot be guaranteed. Please note that resubmissions must be submitted within six months of the date of this email. In exceptional circumstances, extensions may be possible if agreed with the Editorial Office. Manuscripts submitted after this date will be automatically rejected.

Please find below the comments made by the referees, not including confidential reports to the Editor, which I hope you will find useful. If you do choose to resubmit your manuscript, please upload a 'response to referees' document including details of how you have responded to the comments, and the adjustments you have made.

To upload a resubmitted manuscript, log into <http://mc.manuscriptcentral.com/rsob> and enter your Author Centre, where you will find your manuscript title listed under "Manuscripts with Decisions." Under "Actions," click on "Create a Resubmission." Please be sure to indicate in your cover letter that it is a resubmission, and supply the previous reference number.

Sincerely,
The Open Biology Team
mailto: openbiology@royalsociety.org

Associate Editor, Professor Rosemary Akhurst

Comments to Author:

In view of both reviewers suggesting major changes and additions to the manuscript, and the statement by one that the data is scientifically unsound and by both of missing methodology and/or algorithms, we cannot accept the paper in its present form. However, if the authors can address all the deficiencies raised, OB would consider a resubmission.

Reviewer(s)' Comments to Author(s):

Referee: 1

Comments to the Author(s)

In general I would like to congratulate you on a clear and well written paper. I do have some minor suggestions which I would like to see you address prior to publication:

1. I found it a long read to get to the actual methods of the simulation and I would have preferred to have seen that after a brief introduction and survey. Forward referencing in the first sections to results obtained from the simulations only became clear after the second read. You should consider an overview of the method in the introduction.
2. In particular, and understandably, you focus on the fitness landscape mechanism to simulate perturbations in the TME. It would be worthwhile to clarify at the start of the methods section the steps involved in the simulation growth so that it is clear how the genetic algorithm works to produce the 3D tumor cell structures. I felt that it would be instructive to lay out the steps of how the simulation uses the fitness function $w(z)$ to produce the cell populations with a worked example.
3. The model is based on the somatic mutation model of cancer, and some recent work (Reeves, M. Q., Kandyba, E., Harris, S., Del Rosario, R., & Balmain, A. (2018). Multicolour lineage tracing reveals clonal dynamics of squamous carcinoma evolution from initiation to metastasis. *Nature Cell Biology*, 20(6), 699–709. <https://doi.org/10.1038/s41556-018-0109-0>) has highlighted the role of higher order structures, specifically cell to cell interactions. It would be interesting in

the discussion section to get your views on how your model could be extended to simulate such models of cancer growth.

4. The proposition regarding the dependence of the kinetics of the TME and outcomes is fascinating. I would draw your attention to the work that has been done on network evolution (which may not at first sight seem relevant) which indicates that in a dynamic environment sudden changes are undergone by networks in their fundamental morphology (the classic reference is Bianconi, G., & Barabási, A.-L. (2001). Bose-Einstein Condensation in Complex Networks. *Physical Review Letters*, 86(24), 5632–5635.

<https://doi.org/10.1103/PhysRevLett.86.5632>). This approach has been used to model cancer evolution (see Tee, P., & Balmain, A. (2019). Critical behavior of spatial networks as a model of paracrine signaling in tumorigenesis. *Applied Network Science*, 4(1).

<https://doi.org/10.1007/s41109-019-0167-7>) It would be fascinating to investigate whether the physics of critical phenomena could throw further light on the precise conditions upon which the TME cause runaway growth.

I hope you find these suggestions helpful.

Referee: 2

Comments to the Author(s)

Comments attached

Author's Response to Decision Letter for (RSOB-19-0234.R0)

See Appendix A.

RSOB-19-0297.R0

Review form: Reviewer 1

Recommendation

Accept as is

Do you have any ethical concerns with this paper?

No

Comments to the Author

You have addressed all of my comments and I am happy to recommend publication.

Review form: Reviewer 3

Recommendation

Accept with minor revision (please list in comments)

Do you have any ethical concerns with this paper?

No

Comments to the Author

In this manuscript, the authors propose that a changing tumor microenvironment may be contributing to the selection of clones or subclones that may evolve into a tumor. They also suggest that this changing TME selection may explain why tumors may not be more common. The idea is interesting although other explanations for why tumors may be less common than expected exist and supported by good circumstantial evidence (at least).

The work is all theoretical and supported by simulations that out of necessity are arbitrary but do illustrate the points proposed.

I suggest the authors read the manuscript carefully again and revise several sentences that do not seem to make sense or are incomplete.

The value of the manuscript is that it will raise the consciousness level of the TME as an active player in tumor evolution.

Decision letter (RSOB-19-0297.R0)

12-Mar-2020

Dear Dr Jiang,

We are pleased to inform you that your manuscript RSOB-19-0297 entitled "Why is cancer not more common? A changing microenvironment may help to explain why, and suggests strategies for anti-cancer therapy" has been accepted by the Editor for publication in Open Biology. The reviewer(s) have recommended publication, but also suggest some minor revisions to your manuscript. Therefore, we invite you to respond to the reviewer(s)' comments and revise your manuscript.

Please submit the revised version of your manuscript within 7 days. If you do not think you will be able to meet this date please let us know immediately and we can extend this deadline for you.

1) A text file of the manuscript (doc, txt, rtf or tex), including the references, tables (including

captions) and figure captions. Please remove any tracked changes from the text before submission. PDF files are not an accepted format for the "Main Document".

2) A separate electronic file of each figure (tiff, EPS or print-quality PDF preferred). The format should be produced directly from original creation package, or original software format. Please note that PowerPoint files are not accepted.

3) Electronic supplementary material: this should be contained in a separate file from the main text and meet our ESM criteria (see <http://royalsocietypublishing.org/instructions-authors#question5>). All supplementary materials accompanying an accepted article will be treated as in their final form. They will be published alongside the paper on the journal website and posted on the online figshare repository. Files on figshare will be made available approximately one week before the accompanying article so that the supplementary material can be attributed a unique DOI.

Online supplementary material will also carry the title and description provided during submission, so please ensure these are accurate and informative. Note that the Royal Society will not edit or typeset supplementary material and it will be hosted as provided. Please ensure that the supplementary material includes the paper details (authors, title, journal name, article DOI). Your article DOI will be 10.1098/rsob.2016[*last 4 digits of e.g. 10.1098/rsob.20160049*].

4) A media summary: a short non-technical summary (up to 100 words) of the key findings/importance of your manuscript. Please try to write in simple English, avoid jargon, explain the importance of the topic, outline the main implications and describe why this topic is newsworthy.

Images

Data-Sharing

It is a condition of publication that data supporting your paper are made available. Data should be made available either in the electronic supplementary material or through an appropriate repository. Details of how to access data should be included in your paper. Please see <http://royalsocietypublishing.org/site/authors/policy.xhtml#question6> for more details.

Data accessibility section

Sincerely,

The Open Biology Team

<mailto:openbiology@royalsociety.org>

Reviewer(s)' Comments to Author:

Referee: 1

Comments to the Author(s)

You have addressed all of my comments and I am happy to recommend publication.

Referee: 3

Comments to the Author(s)

In this manuscript, the authors propose that a changing tumor microenvironment may be contributing to the selection of clones or subclones that may evolve into a tumor. They also suggest that this changing TME selection may explain why tumors may not be more common. The idea is interesting although other explanations for why tumors may be less common than expected exist and supported by good circumstantial evidence (at least).

The work is all theoretical and supported by simulations that out of necessity are arbitrary but do illustrate the points proposed.

I suggest the authors read the manuscript carefully again and revise several sentences that do not seem to make sense or are incomplete.

The value of the manuscript is that it will raise the consciousness level of the TME as an active player in tumor evolution.

Author's Response to Decision Letter for (RSOB-19-0297.R0)

See Appendix B.

Decision letter (RSOB-19-0297.R1)

25-Mar-2020

Dear Dr Jiang

We are pleased to inform you that your manuscript entitled "Why is cancer not more common? A changing microenvironment may help to explain why, and suggests strategies for anti-cancer therapy" has been accepted by the Editor for publication in Open Biology.

Article processing charge

Please note that the article processing charge is immediately payable. A separate email will be sent out shortly to confirm the charge due. The preferred payment method is by credit card; however, other payment options are available.

Sincerely,

The Open Biology Team
mailto: openbiology@royalsociety.org

Appendix A

Response to referees: Why is cancer not more common? A changing microenvironment may help to explain why, and suggests strategies for anti-cancer therapy (Open Biology RSOB-19-0234)

Here we list all comments or concerns raised by the associate editor and referees (in Italic) and our responses to address them below. There are also some minor text changes in the manuscript for clarity (not listed below).

Associate Editor, Professor Rosemary Akhurst

Comments to Author:

In view of both reviewers suggesting major changes and additions to the manuscript, and the statement by one that the data is scientifically unsound and by both of missing methodology and/or algorithms, we cannot accept the paper in its present form. However, if the authors can address all the deficiencies raised, OB would consider a resubmission.

We thank Prof. Akhurst for this summary. We have addressed all reviewers' comments bellow. We thank the editor and the two anonymous reviewers for improving our manuscript.

Reviewer(s)' Comments to Author(s):

Referee: 1

Comments to the Author(s)

In general I would like to congratulate you on a clear and well written paper.

We thank the reviewer for this positive comment.

I do have some minor suggestions which I would like to see you address prior to publication:

1. I found it a long read to get to the actual methods of the simulation and I would have preferred to have seen that after a brief introduction and survey. Forward referencing in the first sections to results obtained from the simulations only became clear after the second read. You should consider an overview of the method in the introduction.

We thank the reviewer for this suggestion. We added the following text in the Introduction.

“Here, we give an overview of the method. To model spatially evolving cancers using fitness landscapes, we establish a phenotypic and genetic model (see Results and Method for details). Cancer adaptation is modelled by Fisher’s geometric model with random mutation and a changing phenotypic optimum. A transformed tissue stem cell is assumed to acquire a genetic or a phenotypic change that initiates cancer growth with a certain fitness that changes in each cell of the growing cancer with mutations and changing TME Properties of genotypic landscapes of selected driver mutations are characterised by using the concepts of Sewall Wright’s genotypic fitness landscapes.”

2. In particular, and understandably, you focus on the fitness landscape mechanism to simulate perturbations in the TME. It would be worthwhile to clarify at the start of the methods section the steps involved in the simulation growth so that it is clear how the

genetic algorithm works to produce the 3D tumor cell structures. I felt that it would be instructive to lay out the steps of how the simulation uses the fitness function $w(z)$ to produce the cell populations with a worked example.

We added the following text at the start of the Method section to make it clear.

“We now describe how the simulation works to produce the 3D tumour cell structures using the fitness function $w(\mathbf{z},t)$ (equation (2), Supplementary Figures S1-S2). The simulation is fully individual-based, following simple growth mechanics described previously (28, 62). Briefly, the offspring of newly divided cells stochastically look for available positions in a 3D lattice space. The population evolves in a discrete and non-overlapping manner. The empty 3D tumour space represents the tumour microenvironment that may change, following the dynamics described above, and the position in the 3D space does not affect the TME or selection (except in the model extension of cell-TME interaction, see Supplementary Note 3). A computer simulation of cancer evolution then proceeds in the following fashion. At each generation, the initial fitness ($w_0 = w(\mathbf{z}_0)$) of each cancer cell is conferred by a starting phenotype, \mathbf{z}_0 , which is unchanged from the previous generation unless a new mutation (advantageous or deleterious) occurs by chance, or phenotypic plasticity arises, for example from disturbances such as inflammation (15). The phenotypic effect (size) of a mutation is sampled from a multivariate normal distribution. A mutation with large fitness effect can move the cancer cell a long phenotypic distance relative to its phenotypic optimum in the fitness landscape. The changing phenotypic optimum of the TME can also change each cancer cell’s fitness. When the cell phenotype and/or phenotypic optimum changes, the Euclidean distance, d , of the phenotype from the optimum will change to d^c , with an associated higher or lower fitness value $w(d^c)$. All these changes in fitness naturally lead to different levels of selection and adaptation – a cancer cell acquires higher fitness when mutations move its phenotype closer to the optimum. Depending on the fitness effects of the mutations in that cell and position of the phenotypic optimum at generation, t , we assume that every cancer cell has to survive viability selection according to its birth rate (the probability to give daughter cells during viability selection – equivalent to its fitness at generation t), $w(\mathbf{z},t)$ and death rate (the probability of cell death during viability selection), $1 - w(\mathbf{z},t)$. The surviving cancer cells then reproduce asexually. New daughter cells randomly occupy available tissue space in 3D according to pre-specified limits on tumour size that may vary with time. The tumour may go extinct, persist with varying size, or grow continuously until it reaches its maximum allowed space or size (nominally representing clinical presentation), after which it continues to be under viability selection without expanding.”

3. The model is based on the somatic mutation model of cancer, and some recent work (Reeves, M. Q., Kandyba, E., Harris, S., Del Rosario, R., & Balmain, A. (2018). Multicolour lineage tracing reveals clonal dynamics of squamous carcinoma evolution from initiation to metastasis. Nature Cell Biology, 20(6), 699–709. <https://doi.org/10.1038/s41556-018-0109-0>) has highlighted the role of higher order structures, specifically cell to cell interactions. It would be interesting in the discussion section to get your views on how your model could be extended to simulate such models of cancer growth

4. The proposition regarding the dependence of the kinetics of the TME and outcomes is fascinating. I would draw your attention to the work that has been done on network evolution (which may not at first sight seem relevant) which indicates that in a dynamic environment sudden changes are undergone by networks in their fundamental morphology (the classic reference is Bianconi, G., & Barabási, A.-L. (2001). Bose-Einstein Condensation in Complex Networks. *Physical Review Letters*, 86(24), 5632-5635. <https://doi.org/10.1103/PhysRevLett.86.5632>). This approach has been used to model cancer evolution (see Tee, P., & Balmain, A. (2019). Critical behavior of spatial networks as a model of paracrine signaling in tumorigenesis. *Applied Network Science*, 4(1). <https://doi.org/10.1007/s41109-019-0167-7>) It would be fascinating to investigate whether the physics of critical phenomena could throw further light on the precise conditions upon which the TME cause runaway growth.

As reviewer's comments 3 and 4 are related, we address them together. The model suggested has the potential to explicitly model individual cell-cell interactions, which is a different approach compared with our existing results on cell-TME and sub-clonal cell-cell interactions (Supplementary Notes 3 and 5, respectively). We added the following text in the Discussion to indicate how we may integrate this network-based method in our model for explicitly modelling cell-cell interactions in the future.

"To address this, we can readily extend our model to a situation in which a cancer cell's TME is influenced by neighbouring cells with spatial heterogeneity using spatial networks, where tumour evolution could be further modelled in details by graph theories associated with the network (39-41)."

I hope you find these suggestions helpful.

We thank the reviewer for these constructive and helpful comments.

Referee: 2

Xiaowei Jiang and Ian Tomlinson model fluctuations in the tumor micro-environment (TME) to explain why cancer incidence is not higher than what is observed in nature. Hereby TME changes are not explicitly modeled, but their consequence is simulated as a shifting fitness landscape with a single optimum. Adaptation to the changing conditions is modeled as a mutation selection process based on Fisher's classical phenotypic geometric model. The authors conclude that cancer evolution in a changing environment is challenging.

The authors' conclusion implies that changing environments favor homeostasis. It is not clear why this should be the case. Intuitively, the statement that a changing environment reduces incidence of cancer is only true if cancer cells are more specialized or less plastic than normal cells.

Normal tissue homeostasis is essential for the functioning and survival of multicellular organisms and has evolved robust buffering mechanisms to ensure that microenvironmental changes can be accommodated. Cancers may retain a few of these

features, but it is implausible that they are as robust as normal tissues given the cell types they contain and the lack of normal structures (e.g. the intestinal crypt is not recapitulated in a cancer, even if some cancers have structures reminiscent of crypts). There are thus excellent reasons to expect that a changing environment renders cancer cells more vulnerable than normal tissue. It is also important to note that our model assumes that the normal cells are buffered against (or simply not exposed to) the changing TME. There is no notion that normal cells will overgrow to out-complete the cancer - that has almost no biological basis for solid tumours. (We are not sure why the referee suggests that a changing environment would favour homeostasis). We have added the above to the manuscript in Method in response to the reviewer's comments.

The conclusions of the model presented here are vague throughout and do not offer new insights.

We respectfully disagree with the reviewer. There is not a substantial body of literature examining the effects of a changing TME on tumorigenesis. We have deliberately described the results of the modelling in general terms in the main text in the hope of attracting a general readership, and show specific examples of our results in very extensive Supplementary Data.

1. Selective forces facilitating the evolution of cancerous tissues are acting relative to the surrounding normal cells. Yet there is no mention of normal cells and what defines or quantifies the transition to cancer. To show that fluctuating TMEs reduce cancer incidence the authors would need to demonstrate qualitative differences in the fitness function $w(z,t)$, depending on whether z is a normal vs. cancer phenotype. Further they would have to show that $\frac{w(z_{normal},t)}{w(z_{tumor},t)}$ – is by trend larger when environmental fluctuations are high.

We thank the reviewer for pointing this out. For simplicity our model assumes that the normal cells are buffered against (or simply not exposed to) the changing TME. There is no notion that normal cells will overgrow to out-complete the cancer - that has almost no biological basis for solid tumours, although it is rather intriguing as a therapeutic strategy.

Alternatively, what would also support the main take home message are results implying that the variance, the entropy of the system is lower when environmental fluctuations are large: $(w(z | high\ fluctuations)) < \sigma(w(z | low\ fluctuations))$.

We thank the reviewer for this excellent suggestion. We added the new results which compare the effects of low and high (frequency) fluctuations of the TME optimum. With extensive simulations, the new results indeed showed that it is possible the variance of the population fitness could be lower when the environmental fluctuations become large. We added the following text and Supplementary Figure S14.

“Moreover, the variance in population fitness can become lower when the fluctuations of phenotypic optimum in the TME are high (Supplementary Figure S14)”

“Figure S14. Variance in population fitness of cancer adaptation in a cyclically changing TME.

The summary of the simulations shows that the variance of the population fitness can be lower when the phenotypic optimum in the TME has high fluctuations. The period is set at $P = 15$, $P = 90$, $P = 180$, $P = 270$ and $P = 360$, respectively. For each given period P , we simulated five different amplitudes, namely, $A = 0.5$, $A = 2$, $A = 4$, $A = 6$ and $A = 8$, respectively. Error bars are the standard error of the mean (s.e.m.), and 100 independent simulations were performed for each parameter combination. “

If any of the above is mentioned anywhere in the manuscript, it was not evident and needs to be emphasized in the results.

“2. The manuscript is missing data. Especially when the authors mention relevance to treatment. Such statements without data to back them up are not warranted. There is abundant open source data available for the authors to attempt to validate their conclusions.”

We thank the reviewer for this suggestion. Although our study is primarily a theoretical and computational study with qualitative predictions, it would be ideal to test some of the predictions of our modelling results on pan-cancer data, particularly measuring the variable fitness effect of mutations in evolving tumours under changing TME selection dynamics. Unfortunately, we were not able to find longitudinally sampled whole genome sequencing data with sufficient depth, regions, sampling time points and longitudinal TME information under treatment (e.g., changing the phenotypic optimum therapy) or without treatment in solid tumours from clinical settings. In the future we could extend the complexity of the model further to account for some of these deficiencies in the data. We added the above explanation in the Discussion to make this clear.

“In the future our fitness landscape-based spatial modelling could be extended to measure the variable fitness effect of mutations in cancer cells’ adaptation to changing TMEs and treatments, which could use longitudinally sampled whole genome sequencing data with TME information. There has been some success in measuring patient specific sub-clonal selection coefficients by assuming a static TME(50).”

3. Line 307: “Only treatments that induce large optimum shifts can cause a cure”. This statement is vague and intuitively true. There is no new information gained from this conclusion. What constitutes a “large” optimum shift? To arrive to informative conclusions the authors need to consider time, cell counts and toxicity to name a few.

Yes, we did consider time, population size and intensity of the phenotypic optimum shift, etc. By large optimum shift we mean a large sudden change of the phenotypic optimum induced by the treatment (e.g., due to toxicity). The detailed information is in the figure legend of Supplementary Figure 30. We revised this text for clarity.

4. The assumption that fitness landscapes have a single, well-defined optimum is not warranted.

We agree with reviewer that in reality the phenotypic optimum changing pattern may not be well-defined. However, it is a reasonable simplification as we already discussed in the manuscript (e.g., it may be qualitatively similar to aging or hypoxia). The modelling actually follows the multivariate-optimum model, which can change along n-dimensional traits (see e.g., Jones et al. Evolution 2004). For simplicity, we only varied the optimum along a single dimension/trait (see e.g., Matuszewski et al. Evolution 2014). We also added the following text in the Method to make it clear.

“We assume that the TME optimum follows the multivariate-optimum model and only the first trait of the n traits changes(52, 54, 64). It is important to note that the assumption that the fitness landscape has a single, well-defined optimum is a reasonable simplification and assumed qualitatively as suggested before (e.g., due to aging, random tumour-infiltrating immune cells or hypoxia), which could be more complex in reality.”

5. There is value in how broadly the authors varied the dynamics of TME changes. In particular, cyclically changing environments are likely relevant. Examples of naturally occurring periodicities, such as the circadian clock, are worthwhile mentioning. What is missing is a discussion of the different time scales of plausible TME fluctuations relative to the life cycle of cells.

We thank the reviewer for this excellent suggestion. We added the following text in the Discussion.

“Although we showed a cycling phenotypic optimum in the TME, e.g., due to hypoxia, may be particularly capable of promoting adaptive evolution by periodically fixing more driver mutations, the timescales of the plausible TME fluctuations of the optimum and relative life cycle of cancer cells are not clear. Our assumption was that the timescale (the period) of the cycling of the TME is larger than the life cycle of cancer cells. Previous studies have long

showed that the generation time of cell cycles varies considerably among mammalian cells(42). There is also evidence the cycling hypoxia are rather complex with at least two dominant timescales, which vary between hours to days(24). It will be interesting to investigate how the cycling of the TME could be linked to other cell physiologies affecting cancer evolutionary dynamics, such as the circadian clock(43, 44).”

6. Related to the above point: lines 388-391 →The value of the insights gained from the model would increase if the authors were to decide on either one of these processes and focus their modeling approach accordingly.

We did focus on reporting the results of one of the dynamics of TME change, the directionally changing optimum, in the main text for 1) complex 3D spatial-temporal clonal/sub-clonal structures; 2) different patterns of population fitness change; 3) variable fitness effect of driver mutations; 4) different phylodynamics patterns of cancer cell populations, and we put the rest results regarding other optimum changing types and model parameters or extensions to the Supplementary Notes 1-7 for a thorough investigation.

7. That the TME has been neglected for modeling is not true. Fluctuations in TME have been modeled extensively.

We thank the reviewer for pointing this out. We did not mean there is no modelling of the TME. To the best of our knowledge, there is no study which models an evolving tumour spatially in 3D using a formal fitness landscape. We made the following changes in the Introduction to make this clear.

“However, due to challenges in modelling the TME explicitly, the role of a changing TME in determining the fitness effect of mutations in spatially evolving cancers has not generally been considered, particularly in 3D(28, 29).”

Appendix B

Response to referees: Why is cancer not more common? A changing microenvironment may help to explain why, and suggests strategies for anti-cancer therapy (Open Biology RSOB-19-0297)

Reviewer(s)' Comments to Author(s):

Referee: 1

Comments to the Author(s)

You have addressed all of my comments and I am happy to recommend publication.

We thank the reviewer for reviewing our manuscript.

Referee: 3

Comments to the Author(s)

In this manuscript, the authors propose that a changing tumor microenvironment may be contributing to the selection of clones or subclones that may evolve into a tumor. They also suggest that this changing TME selection may explain why tumors may not be more common. The idea is interesting although other explanations for why tumors may be less common than expected exist and supported by good circumstantial evidence (at least).

The work is all theoretical and supported by simulations that out of necessity are arbitrary but do illustrate the points proposed.

I suggest the authors read the manuscript carefully again and revise several sentences that do not seem to make sense or are incomplete.

The value of the manuscript is that it will raise the consciousness level of the TME as an active player in tumor evolution.

We thank the reviewer for these positive comments. Please be assured that we regard the changing TME as just one of several possible explanations for why tumours are not more common.

We went through the manuscript again and indeed identified several problematic sentences in the results section, which have been corrected below. There are also minor text changes made for clarity (not listed below).

“Moreover, the variance in population fitness can become lower when the frequency of the cycling TME optimum becomes higher (Supplementary Figure 14).”

“We confirm that although all treatments assumed to be effective are successful in reducing mean population fitness and tumour mass to the brink of extinction, only treatments that induce a large sudden change of the phenotypic optimum can lead to a cure (Supplementary Figure S29-S30, Supplementary Movies S22-S25).”

“Second, with an elevated mutation rate – perhaps caused by the therapy itself – mean population fitness can quickly rebound and extinction is avoided, where positively selected resistance mutations arise during the simulation (Supplementary Figure S31b).”